# Strategic and Operational Levels of CSR Marketing Communication for Sustainable Orientation of a Company: A Case Study from Bangladesh

**Beata Zatwarnicka-Madura** [1]**, Dariusz Siemieniako** [2] **, Ewa Glińska** [2],***and Yauheniya Sazonenka** [2]

[1] Faculty of Management, Rzeszow University of Technology, Al. Powstancow Warszawy 10, 35–959 Rzeszow, Poland; bezat@prz.edu.pl

[2] Faculty of Engineering Management, Bialystok University of Technology, Ojca Tarasiuka 2, 16–001 Kleosin, Poland; d.siemieniako@pb.edu.pl (D.S.); jessicasazonenko@gmail.com (Y.S.)

* Correspondence: ewaglinska2@gmail.com; Tel.: +48-85746-98-02

**Abstract:** Companies' increasing social awareness has led to the development of a corporate social responsibility orientation, whose implementation impacts on their overall marketing communication, both at its strategic and operational levels. The issue of integration of both levels is recognized as a research gap and is thus, the main interest of this article. A company's CSR orientation depends on the context of social problems, specifically in our case, the need for women's empowerment as well as the creation of sustainable (socially and environmentally) workplace conditions in Bangladesh. The aim of the paper is to investigate and propose ways of integrating strategic and operational levels of CSR marketing communication. We applied the case study research method and specifically with the use of exploratory and descriptive methods, which posits this study within the logic of abductive approach, representing a creative and pragmatic process. The results refer, on the one hand, to the proposition of CSR and sustainable orientation of the one-page strategic plan, but on the other to the merging of the commercial and non-commercial activities of Aarong, a Bangladesh based company. Relying on the one-page strategic plan (OPSP) as a tool of marketing communication at the strategic level, we demonstrated the process of creating a video script scenario as a useful tool at the operational level of marketing communication. The proposed approach to building marketing communication around CSR and sustainable orientation makes the marketing communication consistent and clearer to the audience.

**Keywords:** corporate social responsibility; sustainable orientation; women's empowerment; one-page strategic plan; video script scenario; strategic marketing communication; operational marketing communication; Bangladesh

---

## 1. Introduction

A growing number of companies are currently developing their corporate social responsibility (CSR) in response to a variety of social, environmental and economic pressures. The goal of such activities is to send a signal to the different stakeholders: shareholders, employees, investors, consumers, public authorities and NGOs [1]. Companies that implement CSR orientation integrate social, environmental, ethical, human rights and consumer concerns into their core businesses in close collaboration with their stakeholders [2]. CSR can be used in different directions and at different levels, for example contributing to varied solutions such as ecological, social and other issues, from local to global levels [3]. According to Bhardwaj et al. [4], companies' CSR activities do influence consumers'

purchase decision-making, which can be displayed as consumers' increasing purchase intention or enhanced consumers' willingness to pay higher prices for the companies' products and services. According to Tixier [5], the key element of CSR management is communication. Dawkins [6] suggests that an effective communication of corporate responsibility depends on a clear strategy to evaluate opportunities as well as risks to the brand, thus making it suitable in delivering messages to different stakeholder groups. Dahlsrud [7] notices that at the operational level, contemporary businesses have different and rapidly changing expectations from various stakeholders and this presents new challenges in finding new effective CSR management tools to develop and implement a successful business strategy.

Companies can undertake entrepreneurial ventures, incorporating sustainability into their strategy in response to the uncertain nature of the environment in which they operate and thus adapting their strategic orientation accordingly [8]. Studies of a strategic sustainability orientation of a company have been scarce and vague [8]. Also, the integration of strategic and operational levels of CSR in marketing communication can be recognized as a research gap in CSR literature [3,9,10] which we attempt to develop in this paper. It is obvious that the selection of companies' CSR activities depends on the specificity of a given country or a region's problems, where a company is operating [11]. One aspect of developing a company's CSR actions is the issue of women's empowerment. Over the past decade, the growing concern by businesses to develop CSR initiatives has coincided with the emergence of women and girls as public 'faces' of international development [12]. One of the approaches to women's empowerment is through entrepreneurship. Corporate support for female entrepreneurship spans a remarkable range of industries and takes multiple forms [13]. The increase in the visibility of girls as subjects of global governance is the result, at least in some part, of the convergence of agendas between corporations and development institutions.

According to Barletta [14], the key to creating marketing programs that will attract female business is to understand what women value. The same author [14] states that enterprise communication programs ought to consider factors like social values, which stand for different beliefs and attitudes about how people should relate to each other. Differences in the perception of women and men are also very important. Gender specificity should be included in marketing communication for women [14]. Similarly, in other social areas, such as social marketing programs, there is a need to reduce the level of binge alcohol drinking, which as a form of alcohol consumption, is practiced differently by both men and women [15,16]. Women's empowerment, as part of CSR programs, is currently being developed in business practices, for example by companies like Nike, Goldman Sachs and Coca Cola [12].

Research conducted for this article is based on a case study of Aarong, a company located in Bangladesh with specific social problems regarding gender inequalities that serve as a context for exploring the integration of CSR's strategic and operational levels of marketing communication as a theoretical framework. Bangladesh is a country located in the northeastern part of South Asia, and it is one of the world's most densely populated countries. This is a country where traditional dogmas and ways of living are still predominant, besides its high poverty level, low literacy level of the population and its principal involvement in agriculture. Another issue, in respect of which measures are being taken, is gender inequality. Women as the least established members of the society, "are often considered to be financial burdens on their families" [17]. Gender inequality also leads to difficulties in accessing labor market—the employment rate among women is considerably lower than amongst men; although in many cases working women cannot decide on how to spend the money they earn [17].

Thus, the aim of the paper is to explore and propose ways of integrating the strategic and operational levels of CSR marketing communication. Specifically, the aim is to explore the Aarong's main strategic directions related to CSR and based on that, the aim is to create the schema of a one-page strategic plan (OPSP). A video script scenario that serves as an example of an operational communication tool will also be developed.

To achieve the aim of the paper, we applied the case study research method. As an integral part of the case study, we also used other supplementary research methods, like desk research analysis, observation and seven individual shortened interviews with women from Bangladesh.

The paper is based on the abduction research approach, as no hypothesis had been proposed in relation to the research problem of integration of strategic and operational levels of CSR communication. Our reasoning is based on what we first explored and later created.

The paper is structured as follows. In the literature review section, we at first, present the concept of CSR, the one-page strategic plan (OPSP) and CSR marketing communication with regard to strategic and operational planning, and next we present some important findings of previous studies on corporate video. The research results and propositions section follows the section devoted to the research methodology. The final discussion, conclusions and the references come at the end of the paper.

## 2. Literature Review

Generally, it can be stated that CSR is a company's 'philosophy' that influences all its major business activities and is an integral part of the corporate strategy. Many scholars have tried to define the concept of CSR, but a unified definition is still missing [18]. According to the European Commission (EC), CSR can be defined as the responsibility of enterprises for their impact on the society. In other documents by the EC, we can read that CSR is a "concept whereby companies integrate social and environmental concerns in their business operations and in their interaction with their stakeholders on a voluntary basis" [19]. The basic idea of CSR is to investigate how enterprises integrate stakeholder interests with social values to consolidate the relationship between the organization and society [20].

CSR is important for the sustainability, competitiveness, and innovation of enterprises [21]. The definitions presented indicate that the concept of CSR is about integrating the triple bottom line: social, economic, and the environmental dimensions in a multi-stakeholder dialogue on a voluntary basis [11]. The issue of CSR is becoming more and more important to business organizations. This can be ascribed to several reasons, including the fact that CSR is an instrument that improves the image and reputation of the company, and can be used as a way of legitimizing the company's actions [2]. Companies that adapt CSR values are able to attract and retain best talents, brand image and employee morale and hence can develop such intangible assets into strategic advantages [11]. Consistently with these findings, a growing body of academic research attests to the wide range of business benefits that a company can reap from its engagement in CSR [3].

Nowadays, CSR has evolved, however, from voluntary performance to strategic developments integrated into the core business [2]. According to Amaladoss and Manohar [11] CSR is no longer an option for companies today but is rather a strategic driver of businesses. Companies are devoting, more than ever, substantial resources to various social initiatives, ranging from community outreach and environmental protection, to socially responsible business practices [22]. A strategic level of CSR requires adaptation of strategic planning, which can be defined as a methodology used to lead the company' activities, focusing on its long-term objectives, vision and mission and their deployment [23].

It is commonly known currently, that companies have not only increased their social responsibility, but that customers are showing better social awareness and commercial literacy as well. That is why it is important to communicate a corporate strategy to customers, which enables the companies to visualize their strategies in simple and attractive forms. Strategic planning literature has distinguished one of such forms as the one-page strategic plan (OPSP) which is connected to the concept of road-mapping [24,25]. An example of the utilization of such a tool might be one of the multinational concerns [26], which requires the creation of OPSP by its suppliers as it is, generally, the dominant party in the dyadic relationships with suppliers [27,28]. Technology road-mapping, and its many derivatives, has become one of the most widely applied management techniques for supporting innovation and strategy [24]. Road-mapping is a complex long-term planning instrument that enhances the setting of strategic goals and also for the potential of new technologies, products, and services to be estimated [25].

According R. Galvin [29] (p. 803) "A roadmap is an extended look at the future of a chosen field of inquiry, composed of the collective knowledge and imagination of the brightest drivers of change in that field".

As the name indicates, OPSP is a type of representation of a company's strategy that fits one-page size. It includes information about its mission, vision, strategic goals, as well as other essential information, normally in an image format [25]. The other information includes strategic directions of actions and entities that are involved in relationship with the company [25]. OPSP schema or visualization should also contain the appropriate connections between all kinds of elements presented on the schema.

An important component of a corporate strategic plan is the communication strategy [30]. Corporate communication can be defined as: "A management function that offers a framework for the effective coordination of all internal and external communications with the overall purpose of establishing and maintaining favorable reputations with stakeholder groups upon which the organization is dependent" [31] (p. 5). Modern communication channels dedicated to varied stakeholder groups play significant roles in a communication strategy [32]. Stakeholders are not a homogenous group and they can be divided into three levels according to their influence and importance, namely (i) resource base (company's own resources), (ii) industry structure (entities affecting the industry), (iii) socio-political arena (consisting of the company's political and social environment) [33].

Although companies may be active in CSR programs, their efforts may not make any impact on their business unless they make efforts and choose the right way of communicating them to their stakeholders. Therefore, CSR communication has become an important research issue [11], because—according to Kim [34]—most corporations face difficulties regarding what and how to communicate their CSR efforts to stakeholders more effectively. At the same time, increasing societal pressure for corporations to engage in CSR has led to increasing focus on this issue [35].

The importance of the topic is undeniable, given the fact that while there is extensive literature on CSR, the literature on CSR communication is limited [36,37]. Certain studies in this area of CSR and its communication have already been done, but most of them focus on the topic in the United States and Europe, with little emphasis on the Asian subcontinent [11].

CSR communication is defined as 'communication that is designed and distributed by the company itself, concerning its CSR efforts' [38]. Previous research has shown that CSR communication is a complex and challenging process [38].

CSR has introduced new complexities to the marketing mix, especially when it comes to corporate communication strategies [39]. Information about a company—including its CSR efforts—is an important, active means to attract consumers' attention [9]. The goal of CSR communication should be to provide credible information on corporate initiatives to various interested stakeholders. Some authors believe brands should act proactively because these stakeholders mostly wait to be "educated" about their respective sustainability initiatives [18,40].

Gligor-Cimpoieru and Munteanu [37] distinguish two approaches to communicating CSR: traditional and strategic. In the traditional approach, managers are reticent on communicating the CSR initiatives and their results, both inside and outside the business organization. In turn, in the strategic approach, the communication aspect of SR is a very important part of the success of the initiative because it assures mutual benefits for the social cause and the business organization [37]. A proper communication strategy concerning CSR activities has thus become a key factor for CSR as an effective marketing tool to generate positive outcomes from of a company's CSR engagement and to avoid negative attitudes and behaviors regarding the company. Gruber et al. [9] proposed the need for marketing managers to address a series of questions about CSR communication, namely, which initiatives are to be communicated? What the appropriate message is? And what the most credible channels are?

Consequently, one of the important elements that ensures the success of a CSR communication campaign, is choosing the right channel of communication which should be appropriate to the social

cause, the profile of the business organization and the targeted stakeholders in order to maximize the benefits of the CSR program being implemented [37,41].

The development of the media and the enormous progress in reaching out to potential recipients has resulted in an increase in the demand for video content [42]. Videos are a powerful medium for organizations to promote products and services, explain ideas and strategies, communicate with the public and stakeholders, and educate and train employees. Video is also used to communicate CSR content, which seems to be an effective instrument due to the engagement of many senses and that it is more effective in reaching out to stakeholders [42,43].

Some authors do mention corporate video as a component of Public Relations, which is an element of promotion-mix. Costa Sanches [44] mentions several new audiovisual formats for external communication including corporate video. According to Díaz-Méndez [45], there exist different kind of films with some of them being regarded as institutional soft tools. Corporate video requires writing scripts. The script serves as the blueprint for the construction of the company video. It is the document, which determines whether the video will be a sensational success or merely be mediocre. It details the specifics of the video images. It describes editing effects. It describes what actors will say in dialogues and voiceovers and it indicates what types of audio or sound effects are needed. Since the video literally reflects the image of the corporation, it is expected that the script will need to be approved by higher management, if not the CEO (chief executive officer) [46].

The script can contain storytelling elements. Although each corporate storytelling is different in terms of its objectives, they all have some common underlying elements [47]. These include 1) the need to connect emotionally with their target audiences, 2) the simplicity of the message, since stories do not need to be overly complicated, they need to have the potential to be customizable and memorable, 3) the credibility and transparency, regardless of whether the stories are real or not. This is crucial since users/consumers are very aware of the techniques of advertising and, thus, the stories must not appear to be part of a range of commercial or brand messages, but messages about identity. According to researchers, such as Scardino Salvo [48], the most successful video scripts are those that are short and focused on a well-defined topic.

The creation of a successful company video script deserves a well-structured and focused preparatory elaboration, and should be based on the companies' strategy of marketing communication [49]. It can be assumed that the topic of a video script as a 'one sentence story' is determined by the mission and vision of the company while its content should be about how the targeted environment will look like after the company has achieved its goals. Corporate video and one-page strategic plan (OPSP), as pictured communication forms are very appropriate in the era of social media power to spread information rapidly, or in the other words, they have the shape that is appropriate for the "world of mouth" [50].

## 3. Research Method

The study made use of the case study research method, based on the Aarong Company from Bangladesh. The literature overview showed that case studies are used quite often as exploratory and descriptive methods that lead to generalized suggestions, but do not offer any practical tools or actions for a company [51]. According to Hammersley et al. [52], for example, the case data can lead to the identification of patterns and relationships, creating, extending, or testing of theories. Our paper does not propose or verify any hypothesis since it is based on the abductive approach. Charles Sanders Peirce is considered a pioneer in studies concerning contemporary abduction research [53]. The research logic based on abduction is an approach to knowledge production that occupies the middle ground between induction and deduction [54]. Although abduction represents a creative and pragmatic process, it has rarely been utilized [55,56]. Following the research approach suggested by Séraphin et al. [57], we have chosen to move away from context framework (description) based on the case study, to the concept elaboration (creation of practical marketing activities). This approach was selected because it

would help to achieve the main aim of the paper, namely the exploration and proposition to integrate both the strategic and operational levels of the CSR marketing communication within a company.

The case study research was conducted from February to June 2018. We analyzed the activity of Aarong, its results and current strategy. We used Aarong and BRAC's ((former Building Resources Across Communities), an international development organization based in Bangladesh, is the largest non-governmental development organization in the world, in terms of number of employees as of September 2016) secondary data for the analysis. The information was gathered from the following sources: official websites of Aarong and BRAC, Aarong's annual reports, reports on Aarong's outcomes, sourced from organizations and institutions, such as INSEAD, UNICEF and Asian Marketing Federation. We have, for the needs of this analysis, focused on, Aarong's business model; its mission, vision, strategic goals and strategic directions including its socially-oriented activities; its stakeholders; the communication channels used and finally the level of CSR orientation implementation.

Another aspect of the case study research was the analysis of Aarong's macro-environment to understand better the situation of gender inequalities in Bangladesh. It was used also as an observation method to understand the cultural, social, economic and political factors that impact or are capable of impacting on our research problem. We employed the observation method to analyze macro-environmental factors. Generally, we conducted the secondary data analysis based on the analysis of written materials accessible via the Internet as open resources, which contained information about both Bangladesh macro-environment as well as Aarong's activities and stakeholders. We also explored the official websites of the Bangladeshi government; reports of international organizations, such as UNICEF and the Asian Marketing Federation. Finally, we identified five essential documents to be utilized for the data analysis in this research. The documents which were received via web search included: (i) "Demographic and Health Survey", conducted by the National Institute of Population Research and Training and Ministry of Health and Family Welfare; (ii) "Women and girls in Bangladesh" reported by UNICEF; (iii) "Aarong: Social Enterprise for Bangladesh's Rural Poor", conducted under the supervision of INSEAD; (iv) "Aarong, BRAC's Social Enterprises, and Life: An Interview With Tamara Hasan Abed, Senior Director at BRAC"; and (v) "What we do—Microfinance", the internal analysis of BRAC. Although we utilized individual interviews with Bangladeshi interviewees, who are acquainted with the social, cultural, economic and political situation in Bangladesh, they were only supplementary to the main case study research method, which relied on secondary data concerning Aarong. Seven individual shortened interviews were conducted by our Bangladeshi researcher. (We would like to thank for Yusuf Ibne Towhid for his contribution in conducting primary research, the analysis of Aarong company activities and for his suggestions in creative part of this article). Seven women aged between 30 and 50 from both rural and urban environments were interviewed. The interviewed women had nothing to do with the Aarong company, and were randomly chosen from the public. The different age criterion of choosing interviewees reflected the general lifespan experience and we used it to obtain different perspectives on the research problem. The interviewees were chosen on a purposive and convenient basis, as the topic of the interview was of a delicate nature. Two of the interviewees seemed to be more engaged than others in discussing issues related to social problems in Bangladesh. The interviews were unstructured and they included topics relating to the social, cultural, economic and political situation in Bangladesh with a special focus, however, on gender inequality in the country (including the place of women in society), labor market opportunities and working conditions for women, including their salaries. The scholarly emphasis on the value of both types of studies (structured and unstructured) depends on the goal of the researchers. We agree with the opinion that highly structured survey research as well as semi—or unstructured small-n interviews, should each have their fair share in the rigorous scholar's tool kit [58]. The duration of each interview was about 30 min. The length of the interview depended on many factors. The need to conduct short interviews is justifiable, given the relevant circumstances [59,60]. The researcher took notes during and after the interviews, avoiding digital recording, which could influence negatively on

the interviewee's behavior. To enrich our data on the situation of gender inequality in Bangladesh, our Bangladeshi researcher also utilized content analysis from the local media.

After completing the analysis of Aarong's activities and its macro-environment, we utilized such creative methods as brainstorming and storyboarding, to propose OPSP and video script scenario for Aarong, focusing on CSR communication orientation [61]. Brainstorming technique was selected because it creates the environment for generating ideas in a group, allows for ideas to be combined and improved upon, while denying criticism [62,63]. We have also, while creating the OPSP, used the concept of road mapping method, based on Aarong. We followed the example of Cheng et al. [61], who used the road mapping method to create links between factors (e.g., different company's resources) influencing researched phenomena, which were in Cheng et al. [64] work, the company's objectives and activities.

The OPSP schema was used to create a video script scenario. We made use of information from the OPSP to create a message relating to Aarong's potential CSR communication proposition in a form of video script scenario. The scenario was developed using the storyboarding technique, approved by professional movie makers, and which according to Doyle et al. [65] can be used not only in the movie industry. Its visual nature makes storyboarding helpful in monitoring if the detailed contents of video script scenario are consistent with the corporate strategy. In our case, we used it to check the consistency of video script scenario with OPSP, which was created earlier.

## 4. Research Results

### 4.1. The Problems of Aarong's CSR Marketing Communication

The case study research based on secondary data showed that Aarong was founded by BRAC in 1978 as an easy and natural way to empower rural artisans, especially women, to rise above poverty [66]. The company sells handcrafts made by more than 65,000 artisans located all over Bangladesh, 85% of whom are women. The working zones are located in around 2000 villages [67]. In 2012, artisans were able to save 14 million Bangladeshi Taka with support from BRAC microfinance schemes [68]. Available data concerning achievements show that Aarong has continued to grow. For example, increased employment in Aarong (currently 3800 people), increase in retail space (from 1600 sq. ft. in 1978 to 194,100 sq. ft in 2013), increase in sales levels (from almost US $14 million in 2004 to US $62 million in 2015) and expansion of the retail chain. The share of export sales is 5% [69,70].

The diagnosis of Aarong's marketing communication tools, channels and contents, showed that it focuses mainly on commercial messages, but with very limited presentation of messages regarding its CSR orientation. Aarong's dominant attention regarding marketing communication is paid to its commercial essence, mainly advertising the products, whose features are closely connected with the Bangladeshi culture. As a result, the company's promotion messages are directed mainly to consumers, including distributors. Aarong mainly uses, for this promotion purposes channels such us the social media, e-commerce or Internet targeting advertisement. Aarong participates as a participant in trade fairs and fashion shows to introduce their authentic brand to business partners. The limited communication in respect of their non-commercial orientation is due to the publication of the results of their social activities only on the official websites of Aarong and BRAC and in reports of NGOs that monitor the situation in Bangladesh. Such a use of marketing communication does not impact enough influence on other important stakeholders such as: NGOs, sponsors, celebrities, government, local communities and current and potential employees. These stakeholders are interested in the company's social orientation, their relevant programs and the results of such actions.

Aarong's current mission statement content is "to empower people and communities in situations of poverty, illiteracy, disease and social injustice. Our interventions aim to achieve large scale, positive changes through economic and social programs that enable men and women to realize their potential" [71]. Their vision, on the other hand, is "to create a world free from all forms of exploitation and discrimination where everyone has the opportunity to realize their potential" [71].

Although Aarong's mission and vision statements content are socially oriented, Aarong's extended programs including women's empowerment orientation or environmentally friendly manufacturing are not emphasized within its current strategic direction as well as in the strategic or operational levels of marketing communication. The high level of sustainability of Aarongs' activity achieved through the merging of its non-commercial and commercial wings, resulted in the creation of a marketing communication opportunity. However, the contents of the mission and vision statements presented above indicate that there is a lack of joint social and commercial orientation, although both areas are strongly interrelated in the company's activities. It can also be concluded that the operational level of Aarong's CSR marketing communication suffers from lack of integration with its strategic level of CSR communication.

### 4.2. CSR Oriented One-Page Strategic Plan Integrated Commercial and Non-Commercial Activities

The one-page strategic plan (OPSP) schema with CSR orientation was elaborated on, based on the case study of Aarong's activity analysis including the external micro and macro environment (Figure 1). The purpose of OPSP visualization is to create a synthesized content and an attractive form of the company's strategy for marketing communication.

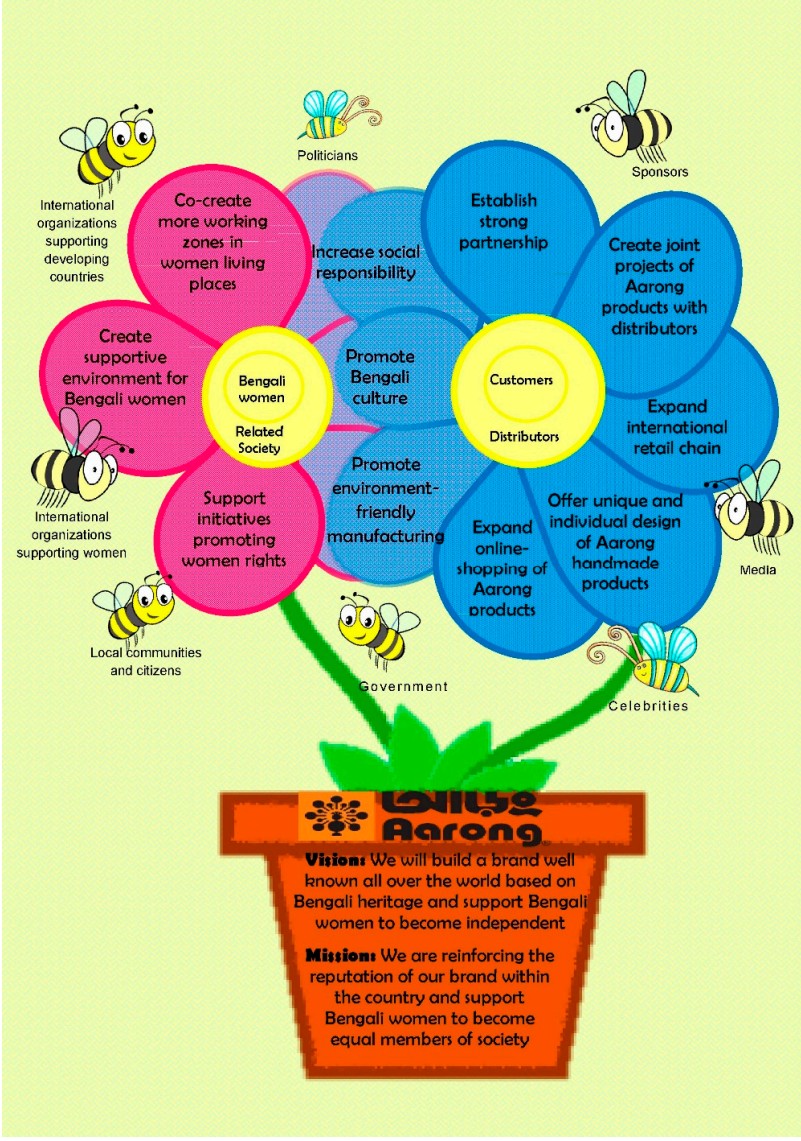

**Figure 1.** CSR oriented one-page strategic plan of Aarong. Source: Authors' own elaboration.

The visual content of OPSP presented in Figure 1, contains key elements of the company's strategic plan, such as its mission and vision, strategic directions, which are general activities for achieving strategic goals and satisfying its different stakeholders. The stakeholders, that are represented in the form of bugs, can be grouped according to the structure proposed by Post et al. [33] such as resource based (Bangladeshi women, customers, sponsors) industry structure (distributors, government), socio-political (local communities and citizens, related society NGOs, politicians, media, celebrities).

The use of a flower in OPSP creation symbolizes the sensitivity and the need to cater for the social problems confronting Bangladeshi women, which has constituted the main concern of Aarong's CSR orientation. Like a flower which grows from a pot, a company's activities are rooted and based on the mission and vision statements, which include associations to Aarong's commercial and non-commercial orientations. The commercial and non-commercial flowers growing from one base symbolize the integration of both of Aarong's orientations. The left side of the flower with the leaves as strategic directions stands for non-commercial activities, while the right side flower stands for commercial activities. The joined leaves of both flowers in the middle of the schema represent some special strategic directions, which are simultaneously related to the company's commercial and non-commercial activities. These include increasing its social responsibility, promoting Bangladeshi culture and, promoting environment-friendly manufacturing. Increasing social responsibility means that Aarong would like their customers, business partners, employees, etc., to not only get financial and relational benefits from cooperation with Aarong, but also to increase their responsibility for solving social problems and engage in social programs, related mainly to women's empowerment supported by Aarong. Promoting Bangladeshi culture means using traditional designs in Aarong's products, which increases public awareness of Bangladeshi culture. Promoting environment-friendly manufacturing is in line with organizing manufacturing in cost-effective ways in their local residences, which as working zones do not produce excess pollution in comparison to a huge factory. The working zones for artists situated in their local environments are solutions applied by Aarong to decentralize manufacturing, so that excessive waists are eliminated, and the workplace fits their familiar environment. Five of the strategic directions distinguished on the right of the pot plant represent the commercial activities only, namely: establishing strong partnership, creating joint projects for Aarong's products together with partners, expanding international retail chain, offering individual and unique design of Aarong's handicrafts, expanding on-line shopping for Aarong's products. These strategic directions mainly aim to achieve their economic goals. In the center of the "commercial flower" the customers and distributors are indicated as entities that focus only on commercial activities, as well as three other kinds of activities that represent joint commercial and non-commercial areas. To the left of the pot plant there are three non-commercial, although strategic, directions distinguished, such as co-creating more working zones for women in their local residences, creating supportive environment for Bangladeshi women, as well as supporting initiatives for promoting women rights. These activities represent the social essence of Aarong's aspirations. Aarong creates working zones in the women's places of residence, so that they could have easier access to work and do not have to abandon their families. Whenever Aarong starts activities in a village, it develops local infrastructure as well, for example, establishing preschools, opening health-care centers, etc. Supporting women to have paid jobs, improving their living conditions and standing for their rights are parts of Aarong's key focus. The non-commercial activities which focus on Bangladeshi women and their immediate society are indicated in the center of the left plant (Figure 1).

### 4.3. Video Script Scenario as a Tool for CSR Marketing Communication—Integration of Strategic and Operational Levels of Communication

Aarong's problems with CSR marketing communication at the operational level are strongly related to the lack of clear indications regarding its strategic level, which are supposed to be its most important messages. The creation of an operational level of CSR marketing communication is supposed to reflect these strategic directions. In order to integrate both levels, while designing the OPSP visualization

(Figure 1), it is necessary to cascade the most important strategic ways to match the operational levels. These most strategic ways are also sources of key information directed at stakeholders, who are the main beneficiaries of such information. When treating women's empowerment as Aarong's key CSR orientation, the video script scenario (VSS) base should relate to the given social problem, namely, the most important strategic directions indicated in OPSP. Those included in our interpretation are: co-creating more working zones for women in their areas of residence, creating supportive environment for Bangladeshi women, promoting Bangladeshi culture, establishing strong partnership, increasing social responsibility and promoting environment-friendly manufacturing. Table 1 is a presentation that link Aarong's strategic directions with fragments of specific content of VSS.

**Table 1.** Strategic directions of OPSP as a base for designing VSS.

| Type of Strategic Direction | OPSP Strategic Directions | Examples of a VSS clips' Description and Texts Content (see also Appendix A) |
|---|---|---|
| NC | Co-create more working zones for women in women residential areas | **Clip description**: "Tani and other women are sitting together and sewing (details of decoration are shown, big pictures of products), talking and smiling" (1:21–1:35) **Text content**: "Around 40 years ago, BRAC realized that the most disadvantaged groups in rural communities were women, and sought for a way that would empower this group while also providing income generation opportunities" (49–1:05) |
| NC | Create supportive environment for Bangladeshi women | **Clip description**: "Aarong's representative is passing the through the village. Houses look neater. Kids play in a kind of a preschool. In the background, there's a house with a medical sign. Women, who see Aarong's representative, smile and wave to her." (2:20–2:30) **Text content**: "With the help of Aarong, working mothers, like Tani have access to day care centers for their children, senior workers receive retirement benefits. Rural workers get various supports, including micro-credits; free schooling for children; subsidized sanitary latrines; health care; as well as legal awareness and support" (2:15–2:35) |
| C and NC | Promote Bangladeshi culture | **Clip description**: "Pictures of Aarong's products, with a name under each of them, i.e., National Bangladeshi Sari, National Bangladeshi Panjabi" (0:03-0:07) "Products are delivered to shops. Customers are coming and looking around, find Aarong's products, look at them, smile, and buys some." (1:49-2:15) **Text content**: "From clay pots to diamond jewelry, and silk and cotton fabrics to brass and leather merchandise, Aarong's vast range of innovative products, backed by a robust supply chain and distribution network, makes Aarong truly a Bangladeshi household brand. (1:21–1:35) |
| C and NC | Increase social responsibility | **Clip description**: "Men in suits are shaking hands with ladies in traditional Bangladeshi clothing" (2:48–2:51) **Text content**: "Nowadays Aarong's goals include both commercial and social components, such as supporting women's empowerment, promoting Bangladeshi culture, and expanding retail chain" (1:11–1:21) |
| C and NC | Promote environment-friendly manufacturing | **Clip description**: "Aarong's representative and Tani sit at the front yard and have a conversation again. It is visible that the village is still a green place, no sign of any factory chimneys or big production buildings." (1:17–1:21) "A car passes through the village with no factories, comes and takes the products." (1:35–1:40) **Text content**: "Aarong uses environment-friendly manufacturing process which does not require the construction of extensive production lines. Many of the products are produced in rural and semi-urban off-site locations by workers who have very little exposure to the final products that are sold at the retail level. (1:35–1:45) |
| C | Establish strong partnership | **Clip description**: "Men in suit are shaking hands with each other" (2:45–2:48) **Text content**: "During its long history Aarong has become more than just a social program. Now Aarong unites, in one network, successful businessmen, efficient distributors and talented women who produce demanded and unique products. We kindly invite you to become a part of Aarong family and help to change the lives of people for better." (2:35–2:54) |

Source: Authors' own elaboration; Legend: C—commercial strategic direction; NC—non-commercial strategic direction.

Bangladeshi women, distributors and customers were selected as key groups of stakeholders that should be included in the video, based on the OPSP. In addition, there is need to illustrate how Aarong influences its immediate society and local community. The main idea of the VSS (Appendix A) was to demonstrate how a company with a CSR orientation contributes to changes in people's lives, the impacts it makes on the natural environment, and its impacts on market success. The key message of the scenario is that Aarong as a socially oriented company empowers rural women in Bangladesh to become equal members of their societies. At the same time, Aarong as a business partner, helps to

achieve goals that are mutually benefiting. Apart from Bangladeshi women, the other target audience of the VSS are distributors, NGOs, and international customers.

The VSS (see in Appendix A) consists of clips description and texts content. It shows the life story of a typical Bangladeshi village woman, named Tani and her life transformations because of her cooperation with Aarong. Tani symbolizes the group of Bangladeshi women who could be influenced by Aarong's social programs. Tani, like many other women, is economically poor, has no paid job and devotes her life to household activities. She is engaged in making different traditional Bangladeshi products for her family. Aarong's representative is the second figure in the VSS. Being engaged in cooperation with Aarong, this person notices the talented woman, Tani and offers her a paid job. Aarong's representative influences changes in Tani's life for the better and is determined to help other women from the village as well. This was the reason, why the support of the man in the suit, who symbolizes the Aarong's business partners, was sought. Together they managed to create working zones for many other women and improved infrastructures in the village, specifically by establishing a preschool and a medical center. Further, the VSS demonstrates that traditional Bangladeshi products made by the women have become popular in the market and business partners are satisfied with the fruitful cooperation and the opportunity to be engaged in social programs. At the end of the VSS, it becomes clear that all the parties—Bangladeshi women, Aarong, the business partners, the local community—are happy with the changes that have taken place due to Aarong's activities.

The VSS also contains general information about Aarong and Bangladesh, emphasizing that this region is mainly rural, economically poor and that the woman's position is vulnerable. Furthermore, it is about, how the main directions of social interventions of BRAC and Aarong have transformed and to be more precise have supported rural regions in their support for women's empowerment. It is also about their roles in promoting Bangladeshi culture and expanding retail networks. The discussion that follows later shows that the production process has been organized in an effective, social and environmentally friendly way. The products are made by detached/ independent artists. The pathway adopted by Aarong in achieving results of CSR orientation follows thus: work is contracted to rural artists without separating them from their traditional surroundings and without organizing production lines in huge factories. Finally, Aarong's achievements, which include social infrastructural development in rural regions and the opportunity given to distributors and other business partners to partake in Aarong's social programs in Bangladesh, are showcased. Since the aim of this VSS is to engage new partners and attract new customers to cooperate with Aarong, a female and a male with Bangladeshi accents were selected for the narrating roles to remind the audience of gender equality as well as demonstrate the national identity.

The story described in VSS is sourced from six strategic directions of OPSP, including two, which are non-commercial, one commercial and three which are blends of both. There is also a visible interrelation between strategic directions. For example, the co-creation of working zones for women is closely related to the promotion of Bangladeshi culture since the artists produce items they already know—traditional products. Consequently, exercising cooperation with the women and the buying of their authentic products has resulted in increased social responsibility. The end of the VSS portrays the image of happy and liberated Bangladeshi women, while the products produced by Aarong become popular with customers all over the world. This illustrates Aarong's vision, namely, "to create a brand well-known all over the world based on Bangladeshi heritage and support Bangladeshi women to become independent". The story told in the VSS demonstrates Aarong's approach to the most important stakeholders, and also how to achieve social and business performance at the same in a sustainable manner for society and natural environment's benefits. Therefore, the VSS as an operational communication tool is clearly connected with the OPSP.

## 5. Discussion

This research contributes to the better understanding of ways of integrating both strategic and operational levels of CSR marketing communication, which is one of the most important conditions

influencing marketing communication [72]. According to Du et al. [3], effective marketing communication of a company's CSR policy depends on a clear corporate strategy, which serves as the base for formulating messages aimed at different groups of stakeholders. Diverse and changeable expectations from various stakeholders pose considerable challenges in finding new and effective tools of CSR management [7] to develop and implement successful communication at both strategic and operational levels [73].

In our case study, we used exploratory and descriptive methods, which posit our orientation on the use of abductive approach [54,56], which has been rarely utilized in CSR research. The abduction procedure represents a creative and pragmatic process, which in our research involved using communication tools at both strategic and operational levels by a company with a CSR orientation. We proposed the integration of both strategic and operational levels of CSR marketing communication that is practiced at Aarong. This was achieved in a three-staged approach, which has the logic of proceeding from the more general to more detailed issues (Figure 2).

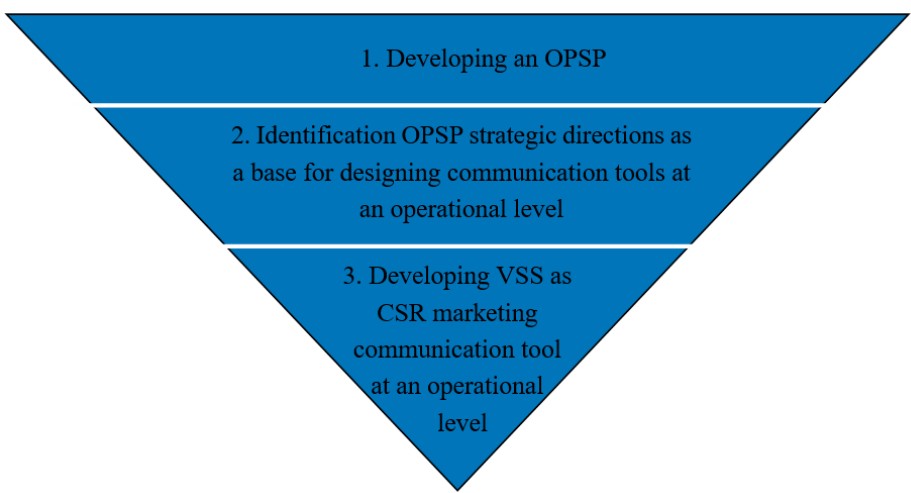

**Figure 2.** A research approach for integrating both strategic and operational levels of CSR marketing communication. Source: Authors' own elaboration.

In developing the one-page strategic plan (OPSP), the general assumption was to rely on Aarong's CSR strategic orientation, a company operating in Bangladesh with its contextual social problems, especially gender inequalities [17]. The integration of both commercial and non-commercial activities in OPSP visualization was proposed in the form of two overlapping pot plants. The merging of commercial and non-commercial activities in a socially responsible business strategy is underlined by Porter and Kramer [74], who emphasized the importance of integrations between a company's strategy and the society on the one hand, and between its competitive advantage and corporate social responsibility on the other. They proposed an analytical model based on the social impacts of the value chain and the role of corporate social responsibility in a competitive context.

The OPSP visualization as a CSR marketing communication at the strategic level, illustrates the integration of Aarong's commercial and non-commercial strategic paths with its CSR orientation. The next step was to ensure the integration of both strategic and operational levels for the needs of CSR marketing communication through the identification of OPSP's strategic paths as the base for designing communication tools at an operational level. Finally, the video script scenario (VSS) was developed as a CSR marketing communication tool at the operational level.

Different aspects of CSR marketing communication are rarely presented in subject literature. Some authors [3] present conceptual framework of CSR communication and analyze its different aspects ranging from message content and communication channels to company- and stakeholder-specific factors that influence CSR communication efficiency. According to these authors, a key challenge in designing effective CSR communication strategy is how stakeholder skepticism can be reduced and how to convey favorable corporate motives in a company's CSR activities. Research was also conducted on consumer

preferences in reference to CSR communication methods [9]. However, aspects of integration of both strategic and operational levels in reference to CSR marketing communication are extremely rare in scientific literature. Engert and Baumgartner [10] did not only emphasize the existence of gaps between corporate sustainability strategy formulation and its implementation in subject literature, but also in respect of CSR communication. Nonetheless, Baumgartner [75] maintains that case studies and other forms of qualitative research approaches can make a valuable contribution to the field.

In our approach, there is some kind of convergence with the approach proposed by Kaplan and Norton [76]. Their comprehensive management model solves one of the greatest management challenges—linking strategy and operations. This powerful six-stage model incorporates the Balanced Scorecard, theme-based strategy maps, with the five Strategy-Focused Organization principles and practices. Kaplan and Norton [76] emphasized developing a strategy using an array of strategy-based tools and planning a strategy using such tools as strategy maps. We opted to employ the tools of strategy maps in our OPSP study. The OPSP format has enhanced our ability to communicate our strategy in a simple and clear way to all stakeholders. Although the literature [77] in the field of strategic and operational management is very rich and provides many different methods, the integration of strategic and operational levels in the aspect of marketing communication is a subject barely noticed. Our theoretical considerations were located in a special context, namely, the choice of a company from Bangladesh, where the specific social problem of gender inequality is commonplace [17]. Gender inequality leads to constraints in accessing labor markets [78]—the employment rate among women is considerably lower than for men [17]. Our propositions for the use of strategic (OPSP) and operational (VSS) marketing communication tools were preceded by the analysis of Aarong's strategic marketing communication tools, channels and contents. Although the content of Aarong's existing mission and vision statements is socially oriented, the extended focus on women's empowerment is neither emphasized in current strategic development paths nor at the strategic or operational levels of marketing communication. The contents of its mission and vision statements do not include the company's joint social and commercial orientation, although, according to our research, both areas are strongly interrelated in the company's activities. The operational level of Aarong's CSR marketing communication suffers from the lack of integration with the strategic level of CSR communication as well. If women's empowerment is to be regarded as a key factor in Aarong's CSR communication message content, it should serve as the base (as a strategic direction signaled in the OPSP) for developing corporate video script scenario and we, thus adopted this approach in the current study. It is also worth paying attention to the aspect of women's empowerment, which has recently been studied from various points of view. For example, the results highlight the important role played by some national and local institutions in women's empowerment [79]

The results of our research have enhanced the understanding of issues of integration of CSR marketing communication at both the strategic and operational levels. The study of Aarong as an example of a truly sustainable company, both in its social [80] and environmental [81] meaning, demonstrated how to develop integrative strategic and operational communication by fusing both commercial and non-profit activities. In the case study of Aarong, we focused on the main instruments of women's empowerment and the most essential message they highlight is that the company has continued to contribute to positive changes in the lives of women, taking care of the natural and social environments, while being market successful at the same time. This issue is particularly important due to the comprehensive approach to management and marketing communication and the inclusion of socially important issues. The significance of the issue is also influenced by the fact of a positive association between CSR strategies and the internationalization of enterprises [82]. Managers should, in the context of a CSR department, be particularly aware of the company's image or reputation in the different markets where it operates [83]. This, on the other hand, constitutes a part of the company's communication policy, which our research has revealed.

Skouloudis et al. [83] presented interesting research results, similar to our findings. Their findings suggest that business professionals highly value such CSR practices that relate to fundamental business

processes and well-established management system standards (i.e. health and safety, environmental management and product quality). Quazi and O'Brien's [84] research results are also interesting in the context of the studies conducted by us. Cluster analysis pointed to two distinctive clusters of managers in both Australia and Bangladesh, with one consisting of managers with a broad contemporary concept of social responsibility, while the other with a narrow view. It was found that corporate social responsibility is two-dimensional and universal in nature and that differing cultural and market settings in which managers operate may have little impact on the ethical perceptions of corporate managers [84].

## 6. Conclusions

The results of our study suggest that managers responsible for CSR strategy of any company should, to a greater degree, consider the integration of both strategic and operational levels in actions focusing on CSR marketing communication. An equally important aspect for managers seems to be care for the harmonization of messages concerning the adopted CSR orientation in marketing communication, both at the stage of strategy planning and its implementation. Taking into account modern research on the role of innovation [85], it is advisable to include innovative ideas in CSR communication.

It is also worthy of note the additional aspect highlighted in the research conducted by Skouloudis and Evangelinos [86]. In their study, an array of issues was explored with the findings emerging from the data sources pointing to multifaceted pressures being exerted on CSR consulting and to their limited role in influencing business conduct as well as to the heightened societal demands for greater corporate responsibility. Nevertheless, it is advisable to take great caution in any comparisons being made due to varying institutional environments and different levels of CSR embeddedness and maturity that exist [86].

This case study has some methodological limitations, such as generalizability of the sampling that establishes boundaries for the interpretation and application of the results. This arises from the fact that the study was conducted for a region with its specific culture, and for a company with a specific business-model. Consequently, cautiousness should be exercised in propagating the results in other regions with different ways of life and well-being and also in companies with different philosophies. The paper does not question if the existing components of Aarong's strategic plan are factual. This study proposed ways of developing communication marketing tools that are capable of demonstrating the CSR orientation of a company [72], in a specific context of social problems, especially gender inequalities.

Another methodological limitation is the fact that no interviews were conducted with Aarong employees, which could be an important and interesting element complementing the case study presented in the current study.

This case study presents only one company with CSR orientation, based on the social enterprise model, located in Bangladesh. Future research that focuses on the integration of strategic and operational levels with reference to CSR marketing communication in different enterprise models would be beneficial as it would allow for comparative analysis in the field of study. As the case study analyzed in this paper is exploratory, future research would also need to concentrate on surveying different companies within a given industry sector and/or region in order to provide knowledge, concerning ways of integrating strategic and operational levels of CSR marketing communication.

Further research should take into account the organizational culture, which may play an important role in CSR and company performance [87,88]. Another issue that needs to be examined is if organizational culture has any impact on the integration of strategic and operational levels of CSR marketing communication.

**Author Contributions:** This article is the result of the joint work by all authors. D.S. supervised and coordinated work on the paper, D.S. and Y.S. conceived, designed, and carried out the methods selection and analyzed the data. E.G and B.Z.-M. have reviewed the literature of the subject, all authors prepared the data visualization, and wrote the paper. All authors discussed and agreed to submit the manuscript.

**Funding:** The research for this paper has been conducted in the framework of projects no. S/WZ/2//2017 and no. DS.ZM.18.001.01 financed from the funds of the Ministry of Science and Higher Education of Poland.

**Conflicts of Interest:** The authors declare no conflict of interest.

## Appendix A

**Table A1.** Aarong's video script scenario.

| Time | Clips Description | Text Content | Lector |
|---|---|---|---|
| 00–03 | Aarong and BRAC logos. | Welcome to Bangladesh—a country with long history and rich cultural heritage. It is located in South Asia. Bangladesh has a population of almost 165.5 million people. Only 36% of Bangladeshi population is urban. In one of the Bangladeshi villages lives Tani. | A man with strong Asian accent |
| 03–07 | Pictures of Aarong products, with their names under each, i.e. National Bangladeshi Sari, National Bangladeshi Panjabi. | | |
| 07–10 | Sun sets over rice fields. | | |
| 10–14 | Poor sunrays fall on the front yard of an old mud house. | | |
| 14–21 | A tired woman is sitting and sewing her son's old clothing. | Tani, like many other women, does not feel herself an equal member of society. She does not have a paid job and she is mainly focused on housework. In Bangladesh, even those women, who work are paid less than their male colleagues. | |
| 21–24 | A tear drops from her eye and runs over her cheek. | | |
| 24–34 | Tani passes through a local market with her kids. All of them are wearing home-made clothes. | | |
| 34–39 | A good-looking woman (Aarong representative) stops them. They have a conversation. Tani looks a bit shy. | The desire to overcome gender inequality and help people like Tani inspired BRAC to start the Aarong project. In the beginning BRAC was mainly focused on empowering the poor, forming a long-term approach to community development. | A lady with strong Asian accent |
| 39–49 | Aarong representative passes through a village. She sees poor houses, kids playing in dirt grounds, women are gossiping. | | |
| 49–53 | Tani and Aarong representative sit in the front yard, drinking water and having a conversation. | Around 40 years ago, BRAC realized that the most disadvantaged groups in rural communities were women, and required a way that would empower this group while also providing income generating opportunities. | |
| 53–1:05 | Tani sews a top (details of decoration are shown), takes it in her hands, looks from all the sides (big pictures of top). She looks a little concerned. She packs and sends the top in a parcel. | | |
| 1:05–1:11 | Aarong representative opens the parcel, looks at the top (big picture of the top), smiles. | Nowadays Aarong's goals include both commercial and social components, such as supporting women's empowerment, promoting Bangladeshi culture, and expanding retail chain. | |
| 1:11–1:17 | Aarong representative comes to an office, shows the top to a person in suit. He thoroughly looks at the top, looks at Nabila with a question and uncertainty on his face. The man nods. | | |
| 1:17–1:21 | Aarong representative and Tani sit at the front yard and having a conversation again. It is visible that the village is still a green place, no sign of a factory chimneys or big production plants. | | |
| 1:21–1:35 | Tani and other women are siting and sewing together (details of decoration are shown, big pictures of products), talking and smiling. | From clay pots to diamond jewelry, and silk and cotton fabrics to brass and leather merchandise, Aarong's vast range of innovative products, backed by a robust supply chain and distribution network, makes Aarong truly a Bangladeshi household brand. | A man with strong Asian accent |
| 1:35–1:40 | A car passes through the village with no production plants, comes and takes the products. | Aarong uses environment-friendly manufacturing process, which does not require construction of big production lines. Many of the products are produced in rural and semi-urban off-site locations by workers who have very little contact with the final products that are sold at the retail level. | |
| 1:40–1:45 | Aarong representative gives Tani an envelope. Tani opens it and sees money. Smiles on her face. | | |
| 1:45–1:49 | Women are sitting together and sewing | Initially, BRAC had a few scattered buyers in Dhaka, with weeks or even months passing between supply and payment. Today, the process is much more streamlined and efficient. Through BRAC's innovative approach, the global community now has a prime example of how targeted mobilization of the poor can support sustainable development efforts, while also generating a financial surplus. | |
| 1:49–2:15 | Days are changing fast. Women are siting, sewing, talking, smiling. Cars are coming, passing through the village with no production plants and taking products. / Products are delivered to shops. Customers are coming and looking around, find Aarong products, look at them, smile, buy. | | |
| 2:15–2:20 | A woman completes a product, looks at it thoroughly from all sides (big pictures of product). / A visitor walks through a shop, finds this product, tries it on, smiles happily (big picture), buys. | With the help of Aarong, working mothers, like Tani have access to day care centers for their children, senior workers receive a retirement benefit. Rural workers get various support, including micro-credits; free schooling for children; subsidized sanitary latrines; health care; as well as legal awareness and support. | |
| 2:20–2:30 | Aarong representative is passing the village. Houses look neater. Kids play in a kind of preschool. In the back ground, there's a house with a medical sign. Women, who see Aarong representative, smile and wave to her. | | |
| 2:30–2:35 | Good looking Tani and Aarong representative sit in the front yard, having a conversation and smiling. | | |
| 2:35–2:42 | Camera moves inside different Aarong Shops in different countries. | During its long history Aarong became more than just a social program. Now Aarong unites in one network successful businessmen, efficient distributors and talented women who produce demanded and unique products. We kindly invite you to become part of Aarong family and help to change the lives of people for better. | A lady with strong Asian accent |
| 2:42–2:45 | Smiling expensive-looking men in suits in offices. | | |
| 2:45–2:48 | Men in suit are shaking hands with each other. | | |
| 2:48–2:51 | Men in suits are shaking hands with ladies in traditional Bangladeshi clothing. | | |
| 2:51–2:54 | Smiling ladies in traditional Bangladeshi clothing, standing in circle, holding hands. | | |
| 2:54–3:00 | Aarong logo, web-site, e-mail, social media details. | For more details, please, visit Aarong 's official web-site. | |

Source: Authors' own elaboration.

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
