# Peer review of "Strategic and Operational Levels of CSR Marketing Communication for Sustainable Orientation of a Company: A Case Study from Bangladesh"

_sustainability, doi:10.3390/su11020555_

Round 1

Reviewer 1 Report

Changes which must be made before publication:
1. The structure of the paper needs attention and the usual rule (introduction-rationale-need for the work/research questions, background-literature review, approach-methods-research performed, results, discussion and then conclusions/concluding remarks) should be followed more closely to facilitate the flow of the paper. Please develop further / expand your discussion of findings perhaps by drawing on relevant studies and in relation with prior MDPI's-Sustainability literature - develop further and expand your final section of concluding remarks; incorporate research and policy recommendations in the final conclusion section. Cite (primarily) in these final-most critical sections of your manuscript relevant papers published in the Journal you submitted your work to (in order to provide some sort of continuity of the specific research string).
2. More references to recent & relevant literature/empirical studies could increase the quality of the research paper and provide a much clearer message to the reader - these may help you building your discussion which needs to be extended. Add the following to your refeence list:

Skouloudis, A., & Evangelinos, K. (2014). Exogenously driven CSR: insights from the consultants' perspective. Business Ethics: A European Review, 23(3), 258-271.

Skouloudis, A., Avlonitis, G. J., Malesios, C., & Evangelinos, K. (2015). Priorities and perceptions of corporate social responsibility: insights from the perspective of Greek business professionals. Management Decision, 53(2), 375-401.

3. The introductory/opening section should communicate a little clearer the literature gaps, as well as the study's aims & objectives in order to facilitate the flow of the study.
4. Concluding remarks – authors must elaborate more on what is their contribution to the literature as well as on opportunities for future research. Questions that need to be answered: Why your study is important? and how it extendso existing knowledge on the issue/topic? Conclusions need to be written in a clear and coherent manner and draw the main lessons from the paper. I suggest you to concentrate on the description of the implications of the work, the main findings and its potential replicability - empirical investigation. Furthermore, limitations of the study need to be outlined to a greater extent, and so are any potential connections between your study and specific aspects of the Journal's scope.
5. Carefully check the references, so as to make sure they are all complete and follow the Guidelines to Authors.
6. Finally, when you submit the corrected version, please do check thoroughly, in order to avoid grammar, syntax or structure/presentation flaws. Make sure you retain a formal/academic-specific style of presenting your work throughout the text - (if necessary) please seek for professional English proofreading services or ask a native English-speaking colleague of yours in order to refine and improve the English in your paper.

Reviewer 2 Report

The paper is interesting, but it is necessary transparently present the methodology.

Reviewer 3 Report

My comments are in attached file 

Round 2

Reviewer 3 Report

My comments are in the attached file. 
